# Adaptive Allocation Algorithm for Multi-Radio Multi-Channel Wireless Mesh Networks

**Walaa Hassan** and **Tamer Farag** *

Department of Management Information Systems, College of Applied Studies and Community Service, Imam Abdulrahman Bin Faisal University, P.O. Box 2435, Dammam 31441, Saudi Arabia; wahassan@iau.edu.sa
*   Correspondence: thhanafy@iau.edu.sa

**Abstract:** The wireless mesh network (WMN) has proven to be a great choice for network communication technology. WMNs are composed of access points (APs) that are installed and communicate with each other through multi-hop wireless networks. One or more of these APs acts as a gateway (GW) to the internet. Hosts of WMNs are stationary or mobile. According to the structure of WMNs, some network features may be affected, such as the overall performance, channel interference, and AP connectivity. In this paper, we propose a new adaptive channel allocation algorithm for a multi-radio multi-channel wireless mesh network. The algorithm is aimed to minimize the number of channel reassignments while maximizing the performance under practical constraints. The algorithm defines a decision function for the channel reassignments. The decision function aims to minimize the traffic around the GW. Whenever the traffic changes in the wireless mesh network, the decision function decides which channel radio reassignment should be done. We demonstrated the effectiveness of our algorithm through extensive simulations using Network Simulator 2 (NS-2).

**Keywords:** adaptive channel allocation; channel allocation; wireless channel reassignment; wireless mesh networks; decision function

---

## 1. Introduction

Over the past decade, the fifth generation (5G) and Internet of Things (IoT) network systems and applications have been developed extensively to deliver a reliable, low latency, real time connectivity network [1]. wireless mesh networks (WMNs) are a part of this technology revelation; therefore, WMN can easily, effectively, and wirelessly connect an entire network using inexpensive, existing technology. The idea of the WMN has risen as an adaptable, flexible, and modest wireless network. The WMN is made of various multiple wireless routers that are distributed in the field, to extend the coverage area of a solitary router that is constrained into a small space. Wireless mesh networks consist of multiple devices typically called an access point (AP), and they are connected to each other. Therefore, WMN is considered as a multi-hop network because each AP can reach another AP by experiencing multiple hops while utilizing intermediate APs as repeaters [2].

At least one AP in a WMN operates as a gateway (GW) to the internet. A host can reach the internet through one of the GWs after connecting to it in multi-hop wireless communications between intermediate APs. Subsequently, wireless links around GWs are normally quite congested, and this may lead to a bottleneck in the entire communications of WMN.

In WMN, each AP performs two distinct roles: one of these roles is connecting hosts to the network, and the other is conveying communications between APs. Therefore, for large scale WMN, it is essential to maintain the performance and decrease traffic toward GWs as the related hosts are increased [3]. To decrease the radio interference between wireless links, we utilize assorted protocols

with various channels (radio frequencies) for them. The IEEE802.11n was appointed as a good choice for WMN communication. IEEE802.11n is a family member of the IEEE802 standard that provides reliability and speed for wireless transmissions. in addition, IEEE802.11n can utilize the unlicensed frequency band 2.4 GHz and unlicensed frequency band 5 GHz. With this protocol, the separation of the roles of APs can be achieved efficiently [4]. The IEEE 802.11ad standard allows for multi-gigabit networking across the 60 GHz range as a way to handle the increase in wireless network traffic for future 5G networks [5].

WMN has many issues to be resolved before being deployed due to the basic network essential requirements for everyday use [6]. For the ideal deployment of WMNs, we examined a few improvement issues and their calculations. In [7], a selection algorithm for an active AP for wireless mesh networks with one GW was introduced. In [8], we characterized the AP allocation issue for adapting the link speed and presented its heuristic algorithm. In [9], we studied the AP allocation for wireless mesh networks with multiple GWs and with different power transmission per wireless links. In [10], we calculated the throughput using IEEE 802.11ac devices for high-speed wireless networks. In [11], we drew a heuristic algorithm for access point allocation in indoor environments with a wireless mesh.

Another important issue that faces WMNs is the channel allocation problem. The efficient utilization of constrained channels to manage the increase in traffic capacity requires the best possible channel assignment. This concept is a crucial issue to achieve a stable WMN. The two methodologies for allocating channels to APs are fixed channel allocation (FCA) and dynamic channel allocation (DCA) [12]. The FCA technique allocates channels for APs in the installation phase with no change over time, as indicated by evaluating the traffic powers among them. The DCA technique may change the direct assignments progressively in real-time, to meet changing traffic forces. In DCA, the channel re-allocations should be as reduced as possible to minimize the holding time of network operations during channel re-allocation procedures. Generally, DCA displays a better capacity and better-delivered performance. This is the motivation for our development of an adaptive channel allocation (ACA) in this paper.

Our objective in this paper is to propose a new adaptive channel allocation algorithm for a multi-radio, multi-channel wireless mesh network with a practical constraint. The practical constraint is on the number of network interface cards (NICs) that can be installed per AP. In this algorithm, we characterize a decision function for the channel re-allocation. Our algorithm is separated into two phases: a static assignment phase and an adaptive assignment phase. In the static (initial) phase, an appropriate, fixed number of network interface cards (NICs) with appropriate channels are assigned to APs, utilizing the connection with the most extreme number of hosts related to each AP, such that the performance is maximized while the maximum number of hosts per AP is minimized. In the adaptive phase, the decision function chooses whether the channel re-allocation must be performed or not, guided by the traffic balance between the links to the GW. The decision function is also intended to minimize the number of channel reassignments to reduce the turn off time of the network tasks during channel re-allocation methods. The effectiveness of our proposed algorithm was verified through Network Simulator 2 (NS-2) simulations. We used our modified 802.11n module on the NS-2 version 2.34 presented in [13].

This paper is organized as follows, In Section 2, we present some of the related work for dynamic channel allocation techniques. Section 3 contains an overview of a mathematical formulation of the adaptive channel allocation problem. In Section 4, we introduce the proposed ACA algorithm. In Section 5, we discuss the simulations and results. Finally, the conclusion is presented in Section 6.

## 2. Related Work

One of the greatest challenges for deploying a sustainable wireless mesh network is the channel allocation problem. The overall performance of the network can be increased by allocating suitable channels to minimize the network interference. One of the greatest challenges for deploying a

sustainable wireless mesh network is the channel allocation problem. The overall performance of the network can be increased by allocating suitable channels to minimize network interference. Wang et al. [14] used partially overlapped channel (POC) based design to improve the network capacity. Their POC algorithm was divided into two steps, neighbor-to-interface binding, and interface-to-channel binding. The first step determined the relation between interfered nodes. The second step determined which channel should be used to minimize the network interference. Although implementing a centralized channel assignment is simple, this puts the burden on the center node. If this node cannot operate properly, the networks will be disrupted.

In [15], the authors proposed a channel assignment scheme based on learning automata. They used a dynamic prediction method to change the channel assignment to improve the network's overall performance. Every node independently calculates a utility value in a stochastic iterative procedure to respond to the currently selected channel. The stochastic procedure is used to solve the problem without giving any information on the optimum action (initially, equal probabilities for all are attached to the proceedings). The allocation of channels varies adaptively with dynamic network conditions by changing the network traffic patterns.

The authors in [16] proposed a method for dynamic channel allocation depending on Bayes' theorem. They aimed to improve the network's throughput and minimize the loss packets by giving the priority of change to the links with the bottleneck. Bayes' theorem is a mathematical method used to measure conditional probabilities. The authors measured the statistical deduction on dropped packets to determine the impact of interference on the crowded links. Therefore, the overloaded links were identified using the traffic capacity.

In [17], the authors introduced communication constraints and measured the overall performance under the usage of partially overlapped channels (POCs). By using a real test-bed, the measured network throughput was improved after using more parallel communications. However, the interference was increased under certain circumstances.

The authors in [18] implemented a channel selection process between the APs base station (BS) and the host stations (STAs). The selection of channels occurs dynamically, in which each STA operates to allocate appropriate communication bandwidth to all STAs and BS regardless of their level of packet generation.

The authors in [19] investigated the channel assignment for unmanned aerial vehicle (UAV) communications using 5G technology. They used millimeter wave (mmWave) bands for cellular services with their applications.

## 3. Mathematical Formulation of Adaptive Channel Allocation Problem

The objectives of this problem aim to maximize the network performance and to minimize the channel re-allocation. In the following section, we introduce the assumptions and mathematical model of the problem under study.

### 3.1. Assumptions

A wireless link may interfere with other wireless links due to the broadcast nature of the links, and this may happen if the links are within transmission range of each other. Interfered links can not transfer data at the same instant of time if they use the same channel. The physical interference model [20] is considered in the current formulation. In real implementation, this is called the carrier-sense multiple access (CSMA)-aware interference model introduced in [21] and implemented in many articles, for example [22]. In this model, both the communication range and the interference range are the same. CSMA is a MAC protocol in which a node verifies the absence of other traffic before transmitting on a shared transmission medium. Essentially, in CSMA, multiple nodes send and receive frames in turn on the same medium without collision unless hidden terminals exist. In this study, we assume that the collision between links within carrier-sensing range are avoided due to

CSMA, and regard that the two directed links interfere with each other only if they are located in the hidden-terminal position.

*3.2. Notations*

Table 1 demonstrates our notations for the mathematical problem formulation.

**Table 1.** Formulation notations.

| Symbol | Definition |
|--------|------------|
| $l_{ij}$ | link between the access point (AP) $AP_i$ and $AP_j$ |
| NOC | non-overlapped channels |
| $F$ | interference matrix between links |
| $t_{ij}$ | expected throughput between $AP_i$ and $AP_j$ |
| $C$ | the matrix of interfered channels |
| $m_i$ | the maximum number of network interface cards (NICs) that can be allowed to $AP_i$ |
| $n_i$ | the output assigned number of NICs to $AP_i$ |
| $ch_{ij}$ | the assigned channel to $l_{ij}$ |
| $FNIC$ | maximum number of hosts per one NIC |
| $Flink$ | total interference of gateway (GW) adjacent links |
| $FStop$ | number of channel re-allocations |

*3.3. Network Mathematical Model*

In this section, we formulate the mathematical model for the network. Suppose that the WMN is described as graph topology $G = (V, E)$. The graph $G$ is a mathematical structure consisting of two sets, $V$ and $L$. The set $V$ is the set of vertices (or nodes) in our WMN. As mentioned before, we shall refer to a node as an AP. Therefore, $V = \{AP_1, AP_2, \cdots, AP_N\}$ where $N$ is the network's total number of APs. Elements of $E$ represent the wireless links between APs; consequently, $E = \{l_{ij} : l_{ij} = (AP_i, AP_j), AP_i, AP_j \in V\}$. One element in $V$ functions as the GW. We suppose that the AP is called $AP_{GW}$. The maximum number of NICs that can be installed in $AP_i$ is $m_i$. The set of non-overlapped channels (*NOC*) in WMN is defined as $C_{NOC} = \{c_1, c_2, \cdots, c_k\}$, where, $k$ is the total number of channels.

Let $F$ be the interference matrix between links.

$$F = \|f_{ijpq}\|, i, j, p, q \in \{1, 2, \cdots, N\}$$

where

$$f_{ijpq} = \begin{cases} 1, & \text{if } l_{ij} \text{ is interfered } l_{pq}, \\ 0, & \text{otherwise.} \end{cases}$$

To utilize the maximum throughput through the network, we calculate the traffic between APs through links, and suppose that $t_{ij}$ is the expected throughput between $AP_i$ and $AP_j$ where $i, j \in \{1, 2, \cdots, N\}$.

In the proposed model, the number of distributed NICs per each AP is calculated through our procedure. Let $n_i$ be the number of NICs calculated by our method for $AP_i$.

Suppose that $C$ is the matrix of interfered channels, such that

$$C = \|C_{ij}\|, i, j \in \{1, 2, \cdots, k\}$$

where

$$c_{ij} = \begin{cases} 1, & \text{if } i = j, \\ 0, & \text{otherwise.} \end{cases}$$

For the standard of the IEEE802.11ax protocol with the orthogonal frequency-division multiplexing (OFDM) [23], a radio signal is divided into smaller sub-signals in the OFDM and placed at different frequencies on separate orthogonal channels. This uses the available spectrum effectively and is less susceptible to interference [24,25].

In this work, we suppose that each AP is associated with the most extreme number of hosts as we designed the network according to the greatest burden for each AP. To compute the total number of NICs appointed to each AP in the whole network, certain critical physical constraints should be applied to the algorithm, and these constraints are as follows:

1. The total number of NICs over the network should not exceed $M = \sum_{i}^{N} m_i$.
2. The output number of assigned NICs for $AP_i$ should be at least one. Suppose it will be $n_i$; therefore, $1 < n_i < m_i$ where $i \in \{1, 2, \cdots, N\}$.
3. Each link $l_{ij}$ should be assigned to a channel $ch_{ij}$, where $ch_{ij} \in \{c_1, c_2, \cdots, c_k\}$.
4. The maximum number of assigned channels is $k$.
5. The chosen channel for the link $l_i j$ at both APs' NIC must be the same $ch_{ij} = ch_{ji}$.
6. The number of channels assigned to the links adjacent to $AP_i$ must be $n_i$ or less.

*3.4. The Outputs*

- The number of NICs assigned to $AP_i$, $n_i$.
- The channel between the two APs $AP_i$ and $AP_j$, $ch_{ij}$.

*3.5. The Problem Formalization*

To achieve a higher throughput and meeting minimum cost, our proposal intends to minimize the following functions: *FNIC*, *Flink*, and *Fstop*, where *FNIC* is the maximum number of hosts per one NIC, *Flink* is the total interference between GW adjacent links, and *Fstop* is the number of channel re-allocations.

$$FNIC = \max_{i} \left\{ \frac{\sum_{j \in G_i} (t_{ij} + t_{ji})}{n_i} \right\}, \tag{1}$$

where $G_i$ represents the set of neighboring APs to $AP_i$.

$$Flink = \sum_{i=1}^{N} \sum_{j=i+1}^{N} \sum_{p=1}^{N} \sum_{q=p+1}^{N} t_{ij} \cdot t_{pq} \cdot f_{ijpq} \cdot c \left( ch_{ij}, ch_{pq} \right), \tag{2}$$

$$Fstop = (Number\ of\ channel\ re-allocations). \tag{3}$$

## 4. Adaptive Channel Allocation Algorithm

Throughout this section, we introduce a method to solve the channel allocation problem dynamically but with adaptive control. The adaptive channel allocation algorithm is separated into two parts, the initial channel allocation (installation phase), and the adaptive channel allocation (running phase).

The initial channel allocation draws out the number of NICs installed per AP in the network and determines the channel allocation as a start up phase. According to this part, the traffic of each AP is assumed to be the maximum associated load.

For the adaptive channel allocation phase, we introduce a decision function to determine whether the network should re-allocate the channel or continue with the last allocation.

*4.1. The Initial Channel Allocation*

Through this section, an initial allocation of the channels is built, considering the case where the network is fully loaded. This phase consists of two stages, the NIC deploying stage and the fixed channel allocation stage to solve the channel allocation problem in the design mode for multi-radio multi-channel WMN. This configuration is considered as FCA, in which the output is a static channel allocation. In wireless mesh networks, the APs topology is fixed; however, APs must collect information regarding the hosts to deploy the initial stage, such information collected by the *hello* protocol as in [26] or other suitable protocols as this is essential to implement the following method.

4.1.1. NIC Deploying Stage

Through this stage, one NIC is deployed to an AP subjected to the maximum traffic load per previously assigned ($t^N IC$) (if this exists) is reached, and all constraints are fulfilled, so that *FNIC* is minimized. This step is repeated for each AP until the maximum load is reached.

1. **NIC Initialization**

   (i) Compute the load per AP from link traffic $t_{ij}$:

   $$t_i^{AP} = \sum_{j \in G_i} (t_{ij} + t_{ji}).\tag{4}$$

   (ii) Allocate one NIC to every AP: $n_i = 1$ as a default.
   (iii) Assign the traffic for each NIC by :

   $$t_i^{NIC} = \frac{t_i^{AP}}{n_i}.\tag{5}$$

   (iv) Set the number the allocated NICs: AN = N.

2. **NIC Deployment**

   (i) Add one NIC to an AP (suppose $AP_s$) where $t_s^{NIC}$ reaches the maximum load while maintaining $m_s < n_s$. After that, $n_s++$ and $AN++$.
   (ii) Stop the phase if $AN = M$ or the physical constraint ($n_s = m_s$) is reached for every AP.
   (iii) Recalculate the traffic :

   $$t_s^{NIC} = \frac{t_s^{AP}}{n_s}.\tag{6}$$

   (iv) Return to step (i).

The Fixed Channel Allocation

The output of the fixed channel allocation phase is a set of channels assigned to the deployed NICs in Section 4.1.1. This phase assigns one channel to one link such that the function *Flink* is minimized according to the following steps.

1. Compute the collision for the traffic:

$$col_{ij} = t_{ij} \cdot \sum_{p=1}^{N} \sum_{q=p+1}^{N} t_{pq} \cdot f_{ijpq}.\tag{7}$$

2. Arrange the links according to the collision in a descending order.
3. Allocate $c_1 \in NOC$ to the most crowded link, let $l_{ij}$ be the first link, and assign $c_1$ to both $AP_i$'s first NIC and $AP_j$'s first NIC.

4.     Allocate the links' channel according to the following filtration steps:

    (a)   *Mandatory channel allocation*:

        (i)     Allocate $c_p \in NOC$ to link $l_{ij}$ if both $AP_i$ and $AP_j$ have NICs allocated to $c_p$. If there are two or more such $c_p$ for both $AP_i$ and $AP_j$, select the channel that minimizes *Flink* without consideration to un-allocated links.

        (ii)    If $AP_i$ has only one NIC allocated with $c_p \in NOC$, and $AP_j$ has no channel, then the link $l_{ij}$ is allocated $c_p$ and one $AP_j$'s NIC allocates to the same channel.

    (b)   *Variety channel allocation*:

        (i)     For each link $l_{ij}$, $AP_i$ and $AP_j$ have both un-allocated NICs, then select $c_p \in NOC$ that minimizes *Flink* without consideration to un-allocated links.

        (ii)    if $AP_i$ and $AP_j$ have two or more NICs in allocated channels $NOC_g \subseteq NOC$, then the link $l_{ij}$ is allocated $c_p \in NOC_g$ to minimize *Flink* without consideration to un-allocated links.

    (c)   *Priority channel allocation change*: If $AP_i$ and $AP_j$ are allocated to $c_p$ and $c_q$ sequentially and $c_p \neq c_q$ and $l_{ij}$ is not an allocated channel, then the channel allocation priority for that link is increased. Return to step 2 after multiplying $col_{ij}$ by a constant number greater than one.

    (d)   *Manage un-utilized NICs*: After the completion of channel allocation for every link in the network, if there are unused NICs, these NICs are switched to other APs with the maintenance of the constraints, and then return to step 2.

*4.2. The Adaptive Channel Allocation*

In this phase, we introduce a new decision function to decide whether to change the channel allocation dynamically or to use the same allocated channel. The concept behind the decision function originates from the limited number of links adjacent to the GW, in which the maximum traffic must pass through those links to explore the internet. The congestion around the GW's links may cause bottlenecks; therefore, the traffic over these links should be distributed evenly to avoid such a situation. The channel re-allocation should occur exactly when the uniformity of the traffic is not fulfilled by the last channel allocation.

4.2.1. Decision Function

The decision function is presented by the following steps.

Step (1)     We begin by calculating the traffic of GW's links, where these links are assigned the same channel $p$, $TGW_p$:

$$TGW_p = \sum_{l_{gi} \in GWL, ch_{gi}=p} t_{gi}, \tag{8}$$

where $GWL$ is the group of links adjacent to the GW, and $ch_{gi}$ is the channel assigned to $l_{gi}$.

Step (2)     We compute the decision factor :

$$\max_{p,q} \left| \frac{TGW_p}{TGW_q} - 1 \right| \geq \lambda, \tag{9}$$

where the parameter $\lambda$ represents the uneven limit of the data transmission utilization for each channel. The parameter $\lambda$ ought to be given experimentally by considering the network topology, the traffic design, and the quantity of NICs distributed to GW. Both $p$ and $q$ are defined as any pair of the channels allocated for links in $GWL$.

The decision function (DF) is defined as follows:

$$DF = \begin{cases} Yes, & \max_{p,q} \left| \frac{TGW_p}{TGW_q} - 1 \right| \geq \lambda, \\ No, & \text{otherwise.} \end{cases} \tag{10}$$

According to the result of the decision function, the channel re-allocation is utilized by using the same steps in Section 4.1, if the value is "Yes"; or if its value is "No", it continues using the last channel allocation.

## 5. Simulation Module Implementation

Throughout this section, we examine our algorithm using our modified NS-2 presented in [13] that supports IEEE802.11n. The simulations are carried out using three different network schemes.

### 5.1. Modified NS-2 Simulator for IEEE802.11n

Network Simulator 2 (NS-2), is an open-source network simulator. We implement modifications to the physical and MAC layers for NS-2 to achieve a simulator for IEEE 802.11n through the framework NS-2.34 Version [27]. The modified model is meant to be proper with the original NS-2.29 for the 802.11 model in [28].

The physical layer of NS-2.34 802.11 is an easy model to understand in which the improved version is embedded in [28]. Through that module, we update the signal-to-noise-ratio to bit-error-ratio (SNR-BER) table to measure the overall network performance and also the error recovery efficiency with various power conditions and multi-input multi-output (MIMO) system schemes. According to that, the new physical wireless files are embedded to enable MIMO to be used as the form of the network interface.

In the MAC layer, we embed the IEEE 802.11n MAC protocol files as a new MAC protocol type for the NS-2.34. We implement only the aggregation MAC protocol data unit aggregation (A-MPDUA) into the MAC layer. Normally, A-MPDUA is less efficient than the aggregation MAC Service Data Unit (A-MSDU) aggregation. However, A-MPDUA can be more efficient in environments with high error rates due to the use of the block acknowledgment mechanism. The acknowledgment of every aggregated data frame is done separately, and to be re-transmitted if suffering from error. In the interface queue type, we implement the new aggregation module. Figure 1 shows our implementations of IEEE 802.11n in NS-2.34.

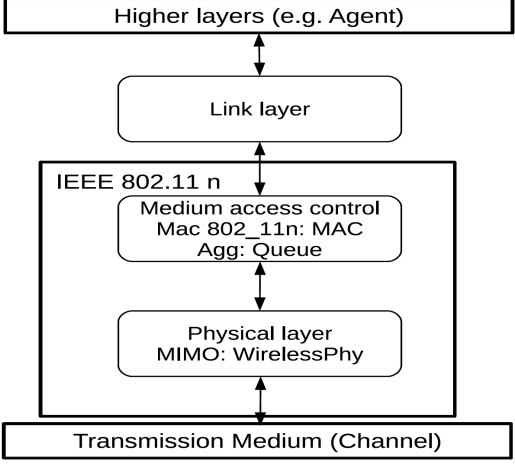

**Figure 1.** IEEE 802.11n implementation in Network Simulator (NS)-2.34.

The simulation is done by assuming that each host has full constant bit rate (CBR) traffic coordinating to the related AP. The length of each data packet handed to the MAC layer is 1000 bytes.

We suppose that the frame size of the MAC layer is a maximum of 8 Kbytes as an aggregated payload. The remainder of the NS-2 simulation assumptions are shown in Table 2.

**Table 2.** NS-2 Parameters.

| | |
|---|---|
| Contention window_Min | 15 |
| Contention window_Max | 1023 |
| Time Slot | 9 µs |
| Short Interframe Space | 16 µs |
| DCF Interframe Space | 34 µs |
| Preamble Length | 16 µs |
| Physical Layer Convergence Protocol Header | 48 bits |
| Physical Layer Convergence Protocol Rate | 6 Mbps |
| Basic Data Rate | 54 Mbps |
| Data Rate | 300 Mbps |

### 5.2. Simulation Instances

Three distinctive schemes are simulated through this work. The first one is the simulation of static channel allocation, in which we assign static channels to the APs after associating the most extreme number of hosts related to each AP. In other words, the channels are fixed regarding any traffic changes.

The second schema is the simulation of a fully dynamic change channel assignment; every time the traffic is changed, the channel reassignment is done. The third schema is the proposed adaptive channel allocation one, in which the decision function decides whether to change the channel assignment or maintain the channel assignments as is.

Through the simulations, we assume that the maximum number of hosts expected to attach to every AP is provided as an input. The hosts are distributed among 24 different times for every AP with random distribution from one of the seven host association patterns, which are illustrated in Figure 2. Each $AP_i$ is loaded with the utmost two or three NICs, $b_i$.

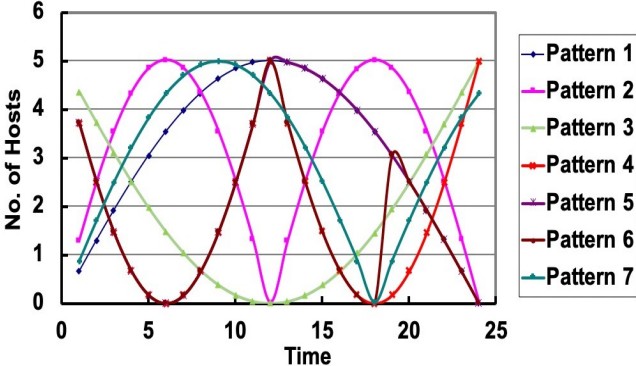

**Figure 2.** Patterns of numbers of the associate hosts, supposing that the maximum number of associated hosts is n.

The simulations are conducted for two network topologies, illustrated in Figure 3. The parameter $\lambda$ relies on the network topology; therefore, the simulations are performed under different values of $\lambda$. Different traffic load patterns are used to summarize the simulations, with the result of the channel assignment numbers.

### 5.3. Network Topology 1 Simulation Results

The first simulation takes place for the network topology 1, $3 \times 3$ ($N = 9$) APs. The whole area of the network is a 300 m $\times$ 300 m field. The distance between APs is 100 m. The GW is selected as shown

in Figure 3. We suppose that the coverage area from AP to host is 100 m. Consequently, the adjoining APs can only communicate with each other. The simulation is carried out through two cases.

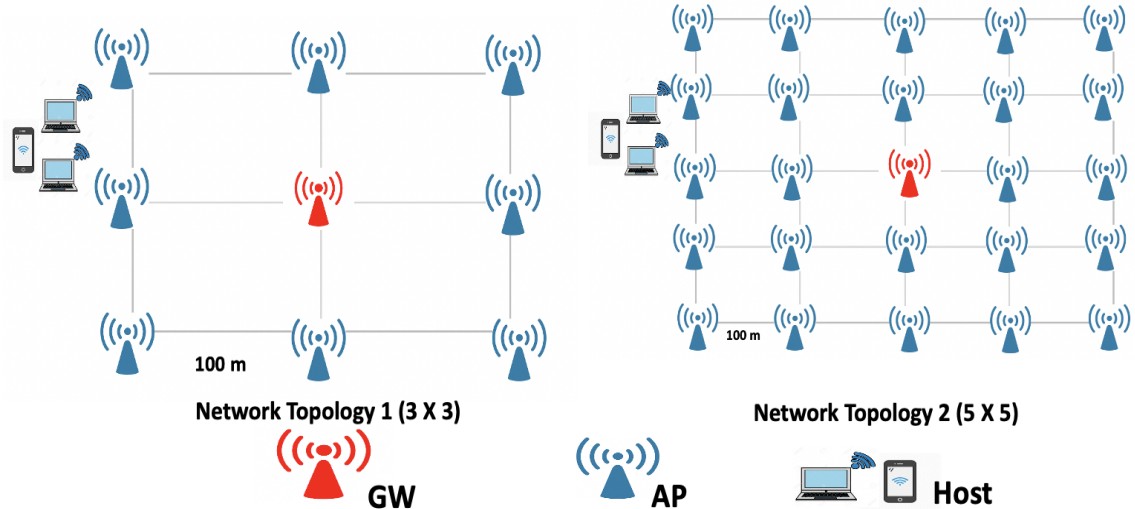

**Figure 3.** Simulation network topologies.

### 5.3.1. Case 1

Figure 4 illustrates the simulation results where two NICs per AP is the maximum number of NICs. Figure 4a indicates that for $\lambda$ values between 0.1 and 0.4, the average throughput of the proposed algorithm is almost as the fully dynamic channel reassignment, which emphasizes the importance of choosing the best $\lambda$. The best-found value of $\lambda$ that achieves the maximum average throughput was 0.1. On the other hand, the relation between the channel reassignment times and the $\lambda$ value is disproportional and, in all instances, is reducing the number of reassignment times.

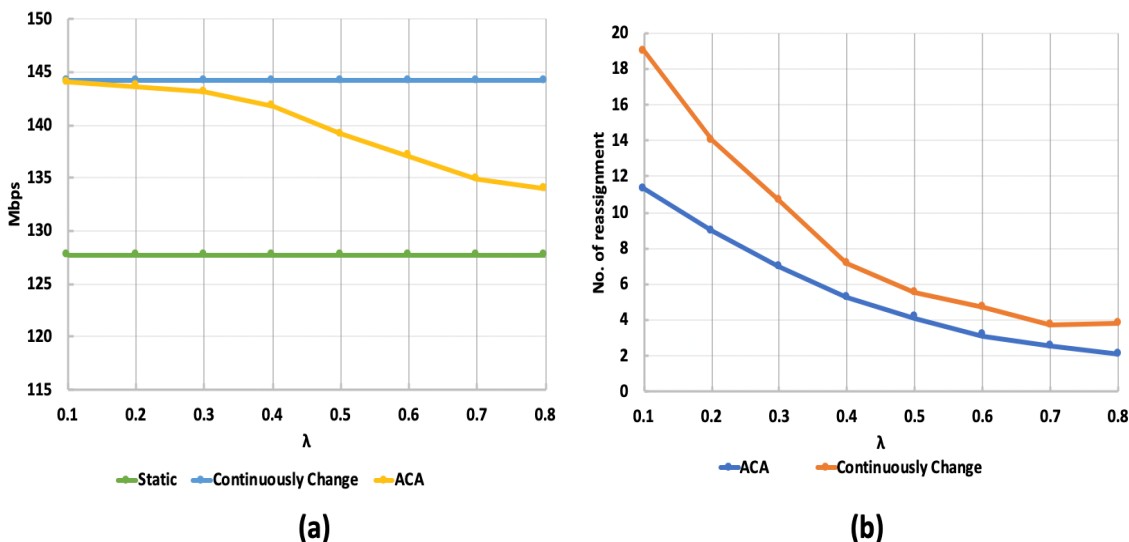

**Figure 4.** The network topology 1 results, the number of channel reassignments, and the overall throughput with the maximum two NICs, (**a**) throughput comparisons and (**b**) channel reassignment comparisons.

For example, concerning the first value of $\lambda = 0.1$, the channel reassignment number is 11.29 over 19 times of dynamic pattern changes as shown in Figure 4b. By choosing $\lambda = 0.8$, we found that the number of channel reassignments is half smaller than the fully dynamic reassignment, and the

measured performance of the proposed algorithm is still superior to that of the static schema. The value of $\lambda$ should be chosen carefully for a real-world network so that the throughput is improved, and the number of channel reassignments is decreased simultaneously.

As a discussion of the illustrated results, we can choose the throughput despite the stopping time caused by the dynamic reassignment. However, this depends on the application that runs over the connected channel, applications, such as voice over internet protocol (VOIP) and video or voice streaming cannot compromise the stopping slots with the average throughput, which demand a need for balancing between the average throughput and the reassignment times.

We can see that the $\lambda$ value of 0.5 is the optimal value to choose between the number of channel reassignments and higher average throughput.

### 5.3.2. Case 2

Figure 5a presents the average throughput for the same $3 \times 3$ network, but with the maximum number of NICs equal to three, while Figure 5b presents the number of channel reassignments for the same case. In these simulations, we found that the ideal value for $\lambda$ was 0.9 to balance between the average throughput and the number of channel reassignments. Clearly, we can say that the $\lambda$ value is a very critical point to obtain the maximum benefits of the proposed adaptive channel allocation. Even with one different parameter, such as the maximum number of NICs per AP, an observable change of the results and the optimal $\lambda$ occurred.

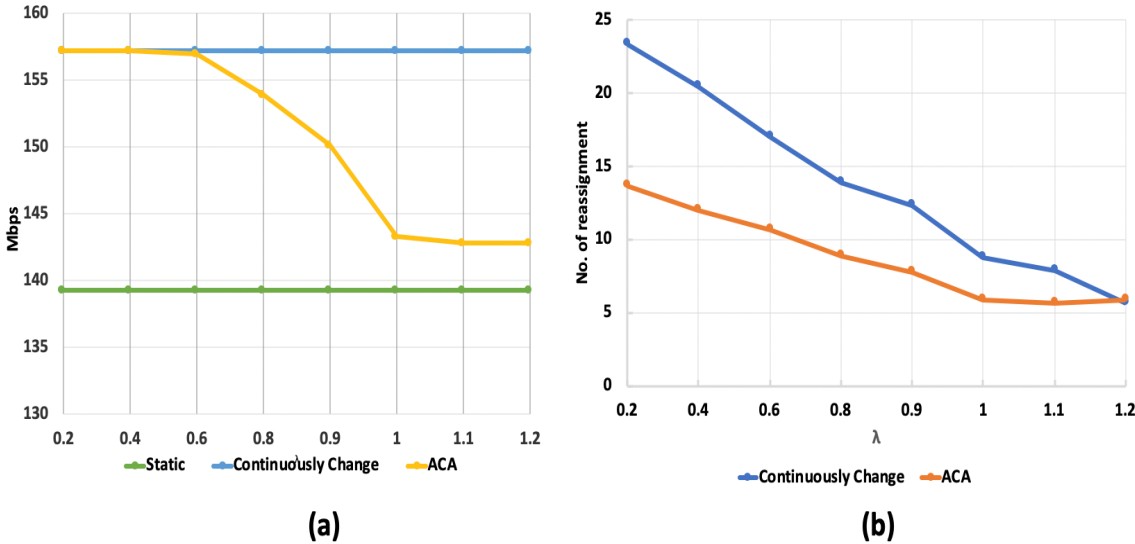

**Figure 5.** The network topology 1 results, the number of channel reassignments, and the overall throughput with the maximum three NICs, (**a**) throughput comparisons and (**b**) channel reassignment comparisons.

### 5.4. Network Topology 2 Simulation Results

The second simulation was performed for network topology 2, $5 \times 5$ ($N = 25$) APs distributed with the 100 m distance apart. Therefore, the whole network is a 500 m $\times$ 500 m field as illustrated in Figure 3. The simulation is conducted through two cases.

### 5.4.1. Case 1

This case is performed using at most two NICs installed per each AP. Figure 6a indicates that, for $\lambda$ values between 0.05 and 0.2, the average throughput of the proposed algorithm is much nearer to the fully dynamic channel reassignment average throughput. The best-found value of $\lambda$ that achieved the best average throughput was 0.05. On the other hand, in Figure 6b we can see that as $\lambda$ increases the number of channel reassignment decreases; however, the best effects of $\lambda$ are for $\lambda$ with

a value 0.1. For example, concerning the first value of $\lambda = 0.05$, the average channel reassignment number is 11.95, over 16.25 times the full dynamic pattern. We found that the best possible scope of $\lambda = 0.25$. By using this value, we can see that the overall performance of the network was enhanced, and the number of channel reassignments decreased and became 5.8 changes over 24 times at the same time. The throughput was dramatically better than the static schema as shown in Figure 6.

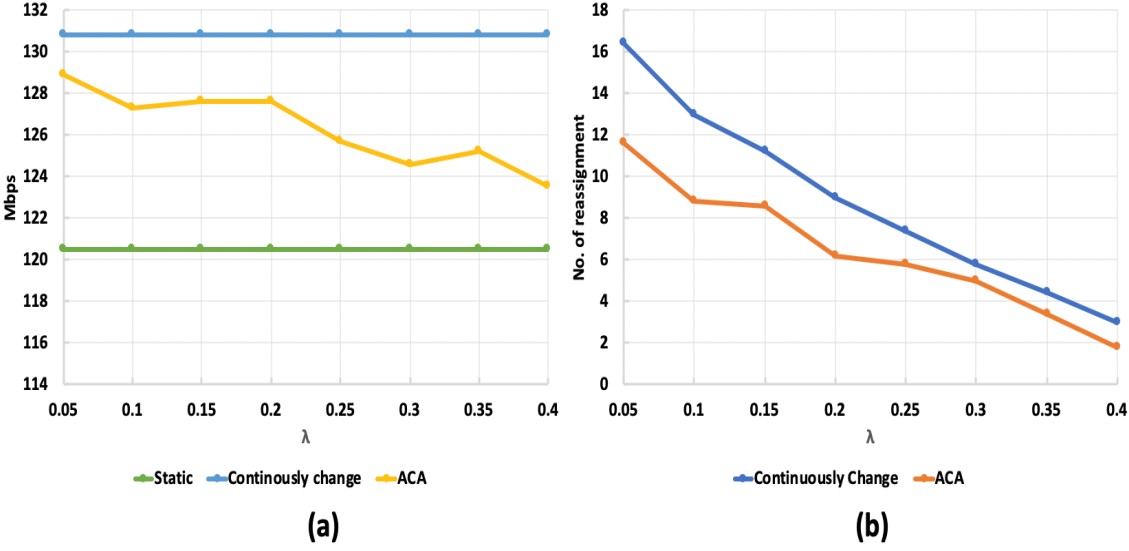

**Figure 6.** The network topology 2 results, the number of channel reassignments, and the overall throughput with the maximum two NICs, (**a**) throughput comparisons and (**b**) channel reassignment comparisons.

### 5.4.2. Case 2

Figure 7 represents the simulations for network topology 2 with three maximum NICs. As noticed, the proper value of $\lambda$ was 0.5 to achieve the balance between the overall performance and the number of channel reassignments. By these simulations, certain values of $\lambda$ may cause worse network performance. As we can see in Figure 7, $\lambda = 0.8$ was the worst value concerning the average throughput and average channel reassignment times. Consequently, we can say that $\lambda$ should be estimated with a mathematical formula to ensure the improvement of the network performance, which will be in future studies. As noticed for all instances, the network throughput with our approach is always better than the static channel allocation. Moreover, the number of networks stop to change the channel is often less than for always changing, which affects the delay of the network.

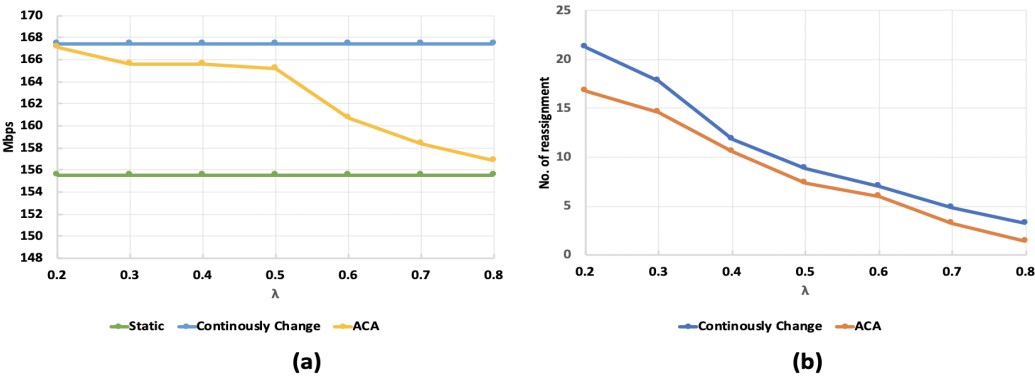

**Figure 7.** The network topology 2 results, the number of channel reassignments, and the overall throughput with the maximum three NICs, (**a**) throughput comparisons and (**b**) channel reassignment comparisons.

## 6. Conclusions and Future Work

In this paper, we presented the formulation of the adaptive allocation problem for multi-radio, multi-channel wireless mesh networks (WMN) with a practical constraint on the number of NICs per AP. The adaptive allocation algorithm is made of the initial phase and the dynamic phase, the dynamic phase introduces a newly defined decision function. The viability of our methodology is checked through our modified version of the NS-2 network simulation. The notable enhancement of the performance is seen by picking a suitable $\lambda$ with a diminishing number of changes. In future works, we shall present a mathematical estimator for $\lambda$, and perform more studies for wireless mesh networks with multiple GWs to justify our methodology.

**Author Contributions:** Conceptualization, W.H. and T.F.; methodology, W.H.; validation, W.H. and T.F.; formal analysis, W.H. and T.F.; investigation, W.H. and T.F.; resources,T.F.; data curation, W.H.; writing—original draft preparation, W.H.; writing—review and editing, T.F.; visualization, W.H. and T.F. All authors have read and agreed to the published version of the manuscript.

**Funding:** This research received no external funding.

**Conflicts of Interest:** The authors declare no conflict of interest.

## Abbreviations

The following abbreviations are used in this manuscript:

FCA    Fixed Channel Allocation
ACA    Adaptive Channel Allocation
GW     Gateway
NIC    Network Interface Card

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
