# Peer review of "Adaptive Allocation Algorithm for Multi-Radio Multi-Channel Wireless Mesh Networks"

_futureinternet, doi:10.3390/fi12080127_

Round 1

Reviewer 1 Report

  • Foremostly the use of English must be improved.  There are numerous typos and many descriptions are not clear.  The descriptions are thus not easily understood.
  • In real deployment, how would the initial loads be made known in the initial NIC deployment stage?
  • In real deployment, interference matrix and matrix of interfered channels are not as clear cut as 1 or 0.  Amount of interference may vary depends on changes to environment.  How do you account for this?
  • From the results it is seen that the smaller the lambda the better the performance.  Please explain why this so and why would other lambda values to be considered.  Is it to reduce number of channel reassignments?  If yes, what is the optimal lambda?
  • Referring to figure 5a, please explain the decrease in performance of "static" and "continuously change" schemes at lambda=1.2.  These 2 schemes should not be dependent on lambda.

Author Response

Dear Prof.

Thank you for your kind revisers through the reviewing process, your comments provided valuable insights to refine its contents and analysis. In this response, we try to address the issues raised as best as possible

Comment 1

  • Foremostly the use of English must be improved.  There are numerous typos and many descriptions are not clear.  The descriptions are thus not easily understood.

Response

To ensure the clarity of the manuscript, we submit it to the English proofing of the system.

Comment 2

  • In a real deployment, how would the initial loads be made known in the initial NIC deployment stage?

Response

Thank you for that comment, it is a challenging question, we suggested in the current version of the manuscript, that the APs uses a discovery protocol called hello protocol to discover their neighbors, then they calculate the maximum load by considering the full capacity links from those hosts. We added a new subsection 4.1 (lines 185--192) to illustrate these steps.

Comment 3

  • In a real deployment, the interference matrix and matrix of interfered channels are not as clear cut as 1 or 0.  The amount of interference may vary depends on changes to the environment.  How do you account for this?

Response

Thank you for that comment, it is a highly important point to be under discussion, in the manuscript we added some lines suggesting the use of CSMA-aware interference model which consider the connections interfered of free for transmission, in the real deployment we suggested that model to defined the interfered links, we chose that model to match our model.

We added a new subsection 3.1 (lines 121--132) to illustrate these steps.

Comment 4

  • From the results, it is seen that the smaller the lambda the better the performance.  Please explain why this so and why would other lambda values to be considered.  Is it to reduce number of channel reassignments?  If yes, what is the optimal lambda?

Response

Thank you for your important spot of lamda value, in the results section we try to emphasize that there are lamda values that cannot improve the performance anymore. We emphasize that the value of lamda is critical in each different network topology and we suggest that we need to find suitable mathematical expression in the near future of that research (section 7). From equation (10) we can say that as lamda reaches near zero the channels around the GW become distributed in balance, which is an indicator of less interference.

Comment 5

  • Referring to figure 5a, please explain the decrease in performance of "static" and "continuously change" schemes at lambda=1.2.  These 2 schemes should not be dependent on lambda.

Response

Thank you for high logical observation in that figure, as observed, there is no relation between the value of lamda and the static or continuously change, we did revise and re-stimulate all of the collected results from the simulator, and it was a miscalculation. The value is the average of the performance form seven different patterns for hosts associations, it has been observed that one of the pattern performance value was not added to the total but divide the result by seven.

The updated figure has been added as shown on page 12.

In the end, thank you very much for that comment that enriches the content of the manuscript and positively in a practical way.

Reviewer 2 Report

Generally speaking, authors of this manuscript have delivered us an very critical and important work from an unique aspect for multi-radio multi-channel Wireless Mesh Networks which is made up of access points (APs) installed and communicated to each other through multi-hop wireless. In this work, a new adaptive channel allocation algorithm for multi-radio multi-channel wireless mesh network is proposed, with the algorithm aimed to minimize the number of channel reassignment while maximizing the performance under practical constraints. 

Although the technical presentation and formulation are solid, sound and easy to follow. Some concerns and advice are recommended as follows in order to enable the paper more interesting to a wider range of readers. 

1. In the so-called 5G era when many 5G systems and applications are being fast and rapidly deployed, it is highly suggested that authors can review more trendy research of relative topic to enrich the introduction and backgrounds. The following references are recommended for authors to consider:
[1] Y. Huo, X. Dong, W. Xu and M. Yuen, "Enabling Multi-Functional 5G and Beyond User Equipment: A Survey and Tutorial," in IEEE Access, vol. 7, pp. 116975-117008, 2019, doi: 10.1109/ACCESS.2019.2936291.

[2] Y. Huo, X. Dong, T. Lu, W. Xu and M. Yuen, "Distributed and Multilayer UAV Networks for Next-Generation Wireless Communication and Power Transfer: A Feasibility Study," in IEEE Internet of Things Journal, vol. 6, no. 4, pp. 7103-7115, Aug. 2019, doi: 10.1109/JIOT.2019.2914414.

2. 11ad/ay mmWave WLAN communications technologies may be also a critical enabling factor, author may consider mentioning them somewhere in the paper.

[3] R. Santos, H. Ogawa, G. K. Tran, K. Sakaguchi and A. Kassler, "Turning the Knobs on OpenFlow-Based Resiliency in mmWave Small Cell Meshed Networks," 2017 IEEE Globecom Workshops (GC Wkshps), Singapore, 2017, pp. 1-5, doi: 10.1109/GLOCOMW.2017.8269214.

Author Response

Dear Prof.

We wish to thank you for your constructive comments in this review. Your comments provided valuable insights to refine its contents and analysis. In this response, we try to address the issues raised as best as possible.

Comment #1:

  1. In the so-called 5G era when many 5G systems and applications are being fast and rapidly deployed, it is highly suggested that authors can review more trendy research of the relative topic to enrich the introduction and backgrounds. The following references are recommended for authors to consider:
    [1] Y. Huo, X. Dong, W. Xu and M. Yuen, "Enabling Multi-Functional 5G and Beyond User Equipment: A Survey and Tutorial," in IEEE Access, vol. 7, pp. 116975-117008, 2019, doi: 10.1109/ACCESS.2019.2936291.

[2] Y. Huo, X. Dong, T. Lu, W. Xu and M. Yuen, "Distributed and Multilayer UAV Networks for Next-Generation Wireless Communication and Power Transfer: A Feasibility Study," in IEEE Internet of Things Journal, vol. 6, no. 4, pp. 7103-7115, Aug. 2019, doi: 10.1109/JIOT.2019.2914414.

Response:

Thank you for your valuable recommendation.

Wireless mesh networks, an emerging technology, may bring the dream of a seamlessly connected world into reality with its applications. Recently, most of these applications deal with 5G communications because of its speed and reliability. Many applications which been deployed using 5G communications are developed on mesh network structure. Your recommended references are added to the introduction section and the related work since they are very related to channel allocation in mesh networks which enabling 5G with user equipment [1].

The added parts are described as follows: in the introduction, (lines 18,19,20).

On the other hand, we will consider the results from [2] in our future work since it is dealing with transmission power for 5G in air-to-ground and air-to-air communications, which is a wireless mesh network structure but communication by 802.11ah. According to your recommendation and advice, we include this topic in our introduction in the third paragraph of the introduction. (lines  40,41,42).

Comment #2:

  1. 11ad/ay mmWave WLAN communications technologies may be also a critical enabling factor, author may consider mentioning them somewhere in the paper.

[3] R. Santos, H. Ogawa, G. K. Tran, K. Sakaguchi and A. Kassler, "Turning the Knobs on OpenFlow-Based Resiliency in mmWave Small Cell Meshed Networks," 2017 IEEE Globecom Workshops (GC Wkshps), Singapore, 2017, pp. 1-5, doi: 10.1109/GLOCOMW.2017.8269214.

Response:

Thank you for your valuable comment

To advance our research, we should enter the 5G era and dealing with millimeter-wave (mmWave) to achieve a fast and reliable and well-performed network.

For that, we add [2] in the related work section as a reference, and we plane to study our algorithm using the IEEE802.11 ad mmWave. We have added it in lines (114,115,116).

In the end, thank you very much for that comment that enriches the content of the manuscript and positively in a practical way.

With great respect